# Epithelial to Mesenchymal Transition History: From Embryonic Development to Cancers

**DOI:** 10.3390/biom11060782

**Published:** 2021-05-22

**Authors:** Camille Lachat, Paul Peixoto, Eric Hervouet

**Affiliations:** 1UMR 1098 RIGHT, University Bourgogne-Franche-Comté, INSERM, EFS-BFC, F-25000 Besançon, France; paul.peixoto@univ-fcomte.fr (P.P.); eric.hervouet@univ-fcomte.fr (E.H.); 2EPIgenetics and GENe EXPression Technical Platform (EPIGENExp), University Bourgogne Franche-Comté, F-25000 Besançon, France; 3DImaCell Platform, University Bourgogne Franche-Comté, F-25000 Besançon, France

**Keywords:** EMT, development, cancer

## Abstract

Epithelial to mesenchymal transition (EMT) is a process that allows epithelial cells to progressively acquire a reversible mesenchymal phenotype. Here, we recount the main events in the history of EMT. EMT was first studied during embryonic development. Nowadays, it is an important field in cancer research, studied all around the world by more and more scientists, because it was shown that EMT is involved in cancer aggressiveness in many different ways. The main features of EMT’s involvement in embryonic development, fibrosis and cancers are briefly reviewed here.

## 1. Introduction

Epithelial to mesenchymal transition (EMT) is a complicated cellular phenomenon that consists in the acquisition, for a cell, of mesenchymal features in place of epithelial ones. EMT can take place in various physiological and pathological contexts. EMT can be determined by numerous molecular mechanisms. EMT can refer to different phenomena with the following common traits: the loss of epithelial features, such as cell–cell interactions and apico-basal polarity, and the gain of mesenchymal ones such as cytosolic expansions, rear-front polarity, and increased migration/invasion capacity. Due to the heterogeneity of the phenomenon called EMT, it would be more correct to use the term “an EMT” rather than “the EMT”.

EMT is involved in many physiological (embryonic development, wound-healing) and pathological (fibrosis, cancers) processes, and is generally classified according to the extracellular context rather than the molecular mechanisms. Type I EMT concerns embryologic development, type II EMT occurs during wound-healing and fibrosis, and type III EMT is found in cancers.

EMT was first described as epithelial to mesenchymal transformation [1], signifying that the switch from epithelial to mesenchymal status was definitive and that only two cellular states are possible: epithelial or mesenchymal. Nowadays, evidence shows that EMT describes a spectrum of intermediate states between an epithelial and a mesenchymal phenotype [2,3], and that cells can switch amongst these intermediate states following extracellular stimuli in a progressive and reversible manner. Here, we will discuss the history of EMT as a research field, explaining how the definitions of this term have been modified over time and highlighting the main discoveries in the domain. This review will help us to better understand actual studies on EMT by considering its history. Then, we will briefly describe the molecular aspect of EMT and the role of EMT during embryonic development and pathologies, highlighting its roles in cancers.

## 2. The First Steps of Research on EMT

### 2.1. The First Phenotypical Observations of an EMT

Elizabeth Dexter “Betty” Hay (1927–2007, Harvard Medical School) was probably the first to describe EMT and later to use this term. This American cellular and developmental biologist worked first on amphibian limb regeneration, from 1958 [4]. She especially described the dedifferentiation of cartilage cells of embryonic salamander’s limbs, which can participate in new limb formation by re-differentiating. This process resembles a kind of EMT. This work led her to study epithelial cells and embryonic development.

She then started to study the role of the extracellular matrix (ECM) in epithelial cell differentiation, showing that ECM composition (for example, collagen concentration) impacts cornea epithelial cells’ differentiation and the secretion of ECM proteins, such as collagen and glycosaminoglycans (GAG) [5].

Elizabeth D. Hay then worked on embryonic development, using chick embryo models. Thanks to the very accurate descriptions of optical and electronic microscopy images obtained from embryonic tissues, she identified and listed different cellular phenotypes during their development. In 1968, she attended the 18th Hahnemann symposium in Baltimore about epithelial–mesenchymal interactions. During her speech, she described how mesenchymal tissues issued from epithelial cells during the migration of neural crest cells in neural tube formation [6]. Indeed, she was describing EMT before we had named it as such. We can probably consider the 18th Hahnemann symposium as the birthplace of EMT research.

Interestingly, in the 1970s, other teams (the team of Prof. Dr. H.-E. Stegner, University of Hamburg, Germany, and the team of Masao Sekiya, Department of Pathology, Nagaoka Red Cross Hospital, Japan) also reported that epithelial and mesenchymal cells coexist in mixed tumors of the uterus [7,8]. Then, there was a debate to determine if these cells come from the same cancer stem cell, or if mesenchymal cells can be derived directly from epithelial cells. At this time, two teams concluded on a common cancer stem cell origin, thinking it was not possible for epithelial cells to acquire a mesenchymal shape or vice versa. Moreover, they found similarities between cancer cells and stroma endometrial cells, the second of which originate the first, according to them [7,8].

Hay and her team used the term “epithelial to mesenchymal transformation” for the first time in 1982, in a publication describing for the first time adult cells undergoing an EMT. They showed that a culture of chick lens epithelial cells (adult or embryonic) suspended in collagen gels can lead to cytosolic expansion, such as in pseudopods [1]. These cells are then able to move individually in the collagen matrix, and they look like mesenchymal cells [1]. In 1981, R. Dulbecco et al. published an article in *Cell Biology* describing a “cuboid-to-fusiform transition” [9]. They observed that the cuboid epithelial cells of rat mammary tumors (induced by a single intravenous injection of N-nitrosomethylurea) can lead to fusiform cells looking like fibroblasts when cultured. This observation was made of many clones obtained from different rats, and also of cells similarly extracted by another group [10]. In 1982, Swiss and German researchers (team of Karl Illmensee, University of Geneva, Switzerland) showed that the first mesenchymal cells appearing during mice embryogenesis, which come from epithelial cells, lose desmosome and cytokeratin expressions and start to express vimentin [11]. They did not use the EMT term, but instead “rapid change from epithelial to mesenchymal character”.

Hay’s team continued to the work on EMT, starting with morphological studies (cytoskeleton description during EMT) using microscopy alone (optical and electronic). Since 1986, this group has been studying the molecular aspect of EMT, using the Western blotting (WB) technique. She managed to show that chicken lens epithelial cells lose type IV collagen expression (basal lamina specific) and γ-crystallin while expressing type I collagen (connective tissues) [12]. After that, this lab studied EMT in thyroid epithelial cells, which present large cytoskeleton remodeling, including a gain of vimentin expression. Moreover, Hay’s team also observed a loss of thyroglobulin expression, which is a precursor of the thyroid hormone, and highlighted cell dedifferentiation [13]. During the following years, Hay’s lab worked on embryonic development and proposed the “fixed cortex theory” to explain neural crest cells’ migration [14,15]. In a publication in 1990, Hay discussed the importance of cell–matrix interactions for mesenchymal cell migration [16].

Hay’s team next revealed that EMT is involved in another process occurring during embryo development: palatal fusion. Epithelial cells from the medial edge of the embryonic palatal epithelium migrate from each side to allow palatal fusion [17].

### 2.2. The First Mechanistical and Molecular Descriptions of EMT

Since 1990, a few teams have shown the great importance of TGF (transforming growth factor) family proteins (TGF-α; TGF-β1–3) during EMT. Indeed TGF-α expression was reported to lead to a mesenchymal and invasive phenotype in rat prostate cancer cells [18]. In 1991, Potts et al. showed the importance of TGF-β3 in embryonic heart endothelial cells EMT [19], and in 1994, they showed that mammary epithelial cells can undergo EMT following TGF-β treatment [20]. In the following years, thanks to the rapid evolution of the relevant techniques, studies about EMT have become more and more mechanistic. Hay’s team showed the involvement of TGF-β3 during EMT in chick embryonic palatal fusion [21]. Later, they detailed the importance of TGF-β family proteins during EMT in chicken embryonic palatal growth, especially during palatal fusion, which is mediated by EMT, itself governed by TGF-β3, and the involvement of SMAD (mothers against decapentaplegic homolog) proteins during TGF-β-induced EMT [22,23]. In 1999, the lab of Peter Ten Dijke (Ludwig Institute for Cancer Research, Uppsala, Sweden) showed for the first time the involvement of SMAD proteins in EMT induction after TGF-β receptor activation [24]. Finally, Hay’s team worked in 2008 on cancer cell lines, showing that the SNAIL family of EMT-ATFs (epithelial to mesenchymal activated transcription factors) can induce TGF-β3 expression [25]. Concurrently, other teams described EMT during the development or use of cancer cell lines, just as Hay’s team did, but using other models. Thanks to the evolution of the techniques, especially the development of immunolabeling (immunofluorescence, Western blotting), EMT studies then became focused on the biochemical mechanisms involved in this phenomenon. TGF-β was then the first molecule identified to induce EMT in the early 1990s. The main events in EMT’s history are summarized in Figure 1.

Research on EMT has become more popular since 2000. Hits when searching “epithelial to mesenchymal transition” in the PubMed database increased from 30 in 2000 to 182 in 2005, and then to 942 in 2010 and 4975 in 2020 (Figure 2).

Elizabeth Hay and her team wrote in 1995 a review named “An Overviewed of Epithelio–Mesenchymal Transformation”, in which they discussed the molecular mechanisms of EMT induction, the genes regulated by or regulating EMT, and the involvement of EMT in pathologies, especially metastases. Interestingly, they mentioned the possibility that EMT is reversible and its potential implication in therapeutics [26]. In another review article published the same year, they used for the first time the term “epithelial to mesenchymal transition”, they described some genes involved in EMT or the inverse phenomenon, mesenchymal to epithelial transition (MET) [27] (Table 1).

## 3. Main Molecular Aspects of EMT

The growing number of studies focused on EMT that have been described previously contributes to a view of the molecular mechanisms governing EMT. Theses mechanisms have been fully reviewed in the last decade [28,29,30,31]. Here, we will give a short summary of the state-of-the-art knowledge on molecular induction pathways, transcription factors, and RNA interference regulation in EMT.

### 3.1. Molecular Pathways Leading to EMT

EMT can be induced by multiple molecular pathways. Here, we will summarize these pathways and briefly describe the most important, which is mediated by TGF-β.

TGF-β pathways were the first to be described in EMT induction. Nowadays, they are still the most documented [32]. TGF-β receptor can activate several intracellular pathways, such as the canonical SMAD pathway, leading to the expression of EMT-ATFs [33]. Other pathways, such as Rho-GTPase [34,35], PI3K/AKT [36] and MAPK [37,38], are activated by the TGF-β receptor, and can also induce EMT. All these pathways are redundant and can act together or separately, which explains the plurality of EMT phenotypes.

Other pathways, independently of TGF-β, can lead to EMT induction. The TNF-α-mediated NFκB pathway or EGF (epidermal growth factor) pathway may act synergistically with the TGF-β pathways [39,40], and other growth factors such as FGF (fibroblast growth factor) [41], HGF (hepatocyte growth factor) [42], IGF1 (insulin-like growth factor 1) [43], PDGF (platelet-derived growth factor) [44] and VEGF (vascular endothelial growth factor) [45] may act through the activation of PI3K/AKT and MAPK signalization. The Wnt [46], Hedgehog [47] and Notch [48,49] pathways have also been described as EMT inducers in many models.

Finally, in the cancer microenvironment, hypoxia and interleukins (mainly IL-6 and IL-8) also lead to cancer cell EMT (Figure 3).

### 3.2. Transcriptional Regulation of EMT

We have previously described that EMT can be induced by various molecular pathways. These pathways often lead to the activation of the expression of EMT-ATFs. The first EMT-ATFs described were Snail and Slug (later called Snail1 and Snail2) [50,51]. These transcription factors can repress or activate gene transcription in response to EMT induction. It was shown in vivo for the first time by Nieto et al. in 1994 [52] that the blockade of Snail or Slug expression can inhibit EMT. Since then, many studies have reported these observations [53,54].

There are two other main families of EMT-ATFs. The ZEB (Zinc finger E-box-binding homeobox) family was described in 2001 [55], and it includes two members, ZEB1 and ZEB2, which are both activators or repressors of transcription. The Twist family (Twist1 and Twist2) of transcription factors was described in 2004 [56].

All of these transcription factors can directly repress E-cadherin expression during EMT [55,56,57,58]. The loss of E-cadherin expression is a central event in this phenomenon, and it was one of the first molecular events that allowed us to associate EMT with cancers [59,60,61].

### 3.3. Non-Coding RNA Regulation of EMT

EMT is largely regulated by non-coding RNA. Here, we will cite two regulation loops mediated by non-coding RNA that are essential during EMT regulation. First, miR-200 can induce ZEB1/2 and Snail1/2 mRNA degradation, and thus a decrease in their protein levels in cells. Actually, the forced expression of miR-200 is sufficient to blockade TGF-β-induced EMT [62,63,64,65]. Moreover, ZEB and Snail family members are able to repress miR-200 expression, forming a regulatory loop [66].

In the same way, miR-34 can also target Snail family mRNA and Snail1/2, directly decreasing the expression of miR-34 and thus constituting a similar regulatory loop [28].

## 4. EMT during Embryonic Development

EMT was first observed during embryonic development. Here, we report the main steps of development by which an EMT takes place.

### 4.1. Gastrulation

At the beginning of the third week of the development, the embryo consists of an embryonic disk including two layers: the hypoblast (primitive endoderm) and the epiblast (primitive ectoderm) (Figure 4A). Some epiblast cells undergo an EMT and move between the two layers to form a third one: the mesoderm [67,68,69]. This step of the development is called gastrulation (Figure 4B). This third layer extends entirely between the two other layers of the embryonic disk, except in two points: the cloacal membrane and the pharyngeal membrane, which are the first indication of future caudal extremity, and the first indication of the cranial extremity of the alimentary tract, respectively.

### 4.2. Neural Tube Formation

At the end of the third week of embryonic development, the embryo consists of three embryonic layers, as was previously described: the ectoderm, the mesoderm, and the endoderm. The neural plate corresponds to a thick area of the endoderm where cells divide rapidly. The edges of this plate grow to form the neural groove (Figure 5A). The neural crests are located on both sides of the neural groove. The neural crest cells then move to the medial edge, which allows the fusion of the neural crests (Figure 5B). During this fusion, the neural crest cells undergo an EMT, and they scatter in the mesoderm [70,71] (Figure 5C). The neural tube then splits from the ectoderm thanks to the scattering of the neural crest cells (Figure 5D).

### 4.3. Embryonic Palatal Fusion

During the ninth week of embryonic development, the embryonic palatal shelves (one each side) move to meet on the medial edge and fuse. First, the epithelia of both shelves stick to each other to form the medial edge epithelium of the embryonic palate. This epithelium then disappears when its epithelial cells undergo an EMT and scatter in the neighboring mesenchyme [72]. Several researchers have described the apoptosis that may be responsible for the disappearance of the medial edge epithelium of the palate [73,74]; however, others showed that the cells undergoing apoptosis were part of the superficial layer of this epithelium. This superficial layer, by undergoing apoptosis, allows cells from the basal layers to fuse and undergo an EMT [17,22]. Recently, it has been shown that these cells do not scatter, but migrate together in the direction of the oral cavity. These cells may undergo a partial EMT, leading to the maintenance of some cell–cell junctions. The most mesenchymal among these cells can lead the migration of the group of cells [75] (Figure 6). This phenomenon is also observed during the formation of metastasis by cancer cells.

Gastrulation, neural tube formation and embryonic palatal fusion are the main steps of embryonic development in which an EMT is described, but EMT is also involved in additional processes. Indeed, EMT and MET control somite formation [76,77,78] and heart valve formation from the embryonic endocardium [19,79,80].

## 5. EMT in Fibrosis

One of the first description of EMT in fibrosis was made by the team of Eric Neilson in 2002 (Nashville, TN, USA) [81]. Using a kidney fibrosis model, they showed that fibroblast-like cells can appear locally as a result of EMT during fibrosis. Later, the team of Raghu Kalluri (Harvard Medical School, Boston, MA, USA) provided evidence that EMT (or endothelial to mesenchymal transition) also contributes to fibroblast-like cells’ appearance in liver or cardiac fibrosis, using lineage-tracing experiments [82,83]. More recently, Kalluri’s lab suggested that myofibroblasts do not derive from epithelial renal cells during renal fibrosis [84]. However, other works from Kalluri’s team, and Angela Nieto’s lab especially (Instituto de Neurociencias, Sant Joan d’Alacant, Spain), provided new data explaining that renal epithelial cells undergo a partial EMT, which is necessary for the recruitment of bone marrow-derived mesenchymal cells and myofibroblasts involved in fibrosis [85,86,87]. The main fibrosis treatment strategies thus target EMT [28].

EMT involvement during fibrosis has been recently reviewed [28,88,89].

## 6. EMT in Cancers

EMT’s involvement in cancers has been controversial, but there is growing evidence that its role is central in cancer aggressiveness, more so than in tumor development. Aggressiveness is exhibited by many cancer cell properties, such as treatment resistance, immune escape, cancer stem cell formation and metastasis. Indeed, tumor aggressiveness is an important factor in mortality for cancer patients, and that is why EMT is currently an important topic in anticancer treatment research. The involvement of EMT in cancers has been well reviewed [90,91].

For example, the analysis of circulating tumor cells (CTCs) is a good way to assess the importance of EMT in cancers—breast cancer CTCs express both epithelial and mesenchymal genes, showing they undergo EMT at multiple levels [92].

### 6.1. EMT Induction Factors of Cancer Cells

Many extra-cellular factors that can induce EMT are present in the tumor microenvironment. They have two main sources of origin: hypoxia [93] and inflammation [94].

Inflammation at the tumor site is characterized by the presence of immune cells, such as MDSCs (myeloid-derived suppressor cells) and macrophages, which secrete growth factors (TGF-β, HGF [94]) and cytokines (TNF-α (tumor necrosis factor α), IL (Interleukin)-1, IL-6, and IL-8) able to induce an EMT in cancer cells [95,96,97]. Moreover, since the SNAIL1 EMT-ATF can also induce pro-inflammatory cytokine expressions (IL-1, IL-6, and IL-8), this promotes a positive regulation loop [98]. Interestingly, it has been shown in spontaneous skin tumor mouse models that immune infiltration is associated with advanced EMT areas in the tumor [3].

When a tumor is growing, some cancer cells exist transiently in a hypoxia state. It has been shown that the transcription factor linked to hypoxia HIF1-α (hypoxia-inducible factor 1-α) can directly induce the expression of SNAIL1 EMT-ATF [99]. EMT induction due to hypoxia in tumors is also linked to TGF-β pathways [100].

### 6.2. EMT and Anticancer Therapy Resistance

Various teams have shown that EMT can induce resistance to classical anticancer treatments. For example, breast cancer patients resistant to chemotherapies have more CTCs and more mesenchymal cells than women with a good response to treatment [101]. A similar resistance to treatment has been associated with EMT in breast and pancreas cancers [102,103].

At a molecular level, EMT-ATFs can induce the expression of genes involved in DDR (DNA damage response) systems, which could counterbalance the effects of chemo- and radiotherapies [104,105]. SNAIL1 and TWIST EMT-ATFs also lead to an overexpression of ABC (adenosine triphosphate-binding cassette) transporters, which are responsible for multiple chemotherapy resistances. In the breast cancer cell line MCF7, TWIST directly binds to *ABCC4* and *ABCC5* promoters [106]. Finally, in HER2 (human EGF receptor 2)-positive breast cancers, the overexpression of *miR-21* leads to resistance to chemotherapy and immunotherapy targeting HER2 in an EMT-inducing manner [107].

### 6.3. Immune Escape

Immune escape linked to EMT has recently been reviewed by Terry et al. [108]. SNAIL1-expressing mouse melanoma cells, when co-cultured with spleen cells, can induce the proliferation of immunosuppressive T_reg_ (regulatory T cells) cells via TGF-β secretion. The silencing of SNAIL1 in these cells reduces T_reg_ proliferation and invasion of melanoma cells [109].

EMT can also lead to a reduced expression of MHC class 1 (major histocompatibility complex) in cancer cells, and so to a diminution of the formation of immune synapses with cytotoxic T cells or NK (natural killer) cells decreasing cancer cell destruction by the immune system [110,111].

Moreover, EMT can induce an overexpression of PD-L1 (programmed death ligand 1), allowing cancer cells to inactivate immune cells harboring the associated receptor PD1 (programmed death 1). EMT induction in lung cancer cells leads to PD-L1 overexpression, and allows the inactivation of cytotoxic T cells and immune escape [112]. The EMT-ATF ZEB1 is involved in PD-L1 expression in MCF7 cells and a mouse lung adenocarcinoma model [113,114].

### 6.4. Metastasis Formation

Various studies showed the involvement of EMT during metastasis, based on tumor cell xenografts in immunodeficient mice and/or the overexpression of EMT-ATFs [115]. SNAIL1/2 is overexpressed in mouse spontaneous mammary tumors, but not in bone marrow metastasis. These proteins are important for local tissue invasion, but not in metastatic colonization. Inhibiting SNAIL1/2 expression in this model prevents metastasis formation [54,116].

Some pioneers’ studies have shown the importance of cell plasticity during the metastatic cascade, describing that a fine regulation of EMT is required to allow cancer cells to metastasize. In other words, cells need “a good” epithelial or mesenchymal shape at each step of this process to metastasize efficiently [117,118,119]. In a model of spontaneous mammary mouse tumors, a few cells undergo EMT, invade neighboring tissues then blood vessels, and then establish in other organs (lung, liver, and lymph nodes). In the secondary organ, cells can undergo an MET to develop a metastasis [120]. However, the involvement of EMT during metastasis formation is still a controversial field. Two main studies suggest that EMT is not essential for this phenomenon, showing that metastatic cells from mouse breast cancer never express vimentin, and that blocking EMT by the overexpression of miR-200 does not reduce metastatic formation in this model [102]. However, this work showed that there is an enrichment of mesenchymal cells in CTCs, suggesting that these cells could be essential for intravasation and could facilitate the intravasation of epithelial tumor cells by collective migration [3,28]. This could explain why metastasis are only formed by epithelial cells in this model. Then, it was demonstrated that the loss of expression of SNAIL1 or TWIST1 does not prevent metastasis formation in mice pancreatic ductal adenocarcinoma [103]. However, other EMT-ATFs could induce EMT instead of SNAIL1 and TWIST1 in these cancers [28]. Moreover, these two papers highlighted that the authors did not achieve the complete blockade of EMT or track all forms of EMT programs. This confirms that these works are not sufficient to prove that EMT is not required in the metastasis process in these models [121,122].

In a model of spontaneous mammary mice tumors, a partial EMT gives cells with both epithelial and mesenchymal features. This seems to facilitate metastasis development, and promotes the collective migration of cells, while full mesenchymal cells migrate alone [2,3]. In this way, the co-expression of epithelial and mesenchymal markers could indicate a poor prognosis in various cancers (breast, liver, and brain) [123,124,125].

### 6.5. Cancer Stem Cells

CSCs (cancer stem cells) are cancer cells with stem characteristics and high clonogenic potential [2]. CSCs are defined as the source of tumor development and/or being capable of being so. They have been described to express a high level of CD44 and a low level of CD24, or to highly express ALDH1 (aldehyde dehydrogenase 1). These features are generated during EMT in breast cancer cells [115,116,126]. The specific stem cell transcription factors OCT4 and NANOG are also frequently expressed during EMT. Both phenotypes are mechanistically linked [127]. In pancreas cancers, EMT is linked to stem cell properties because ZEB1 represses the expression of miRNAs that normally inhibit the stem cell phenotype [128]. SNAIL1 is also responsible for the symmetric division of colorectal CSCs, increasing their pool [129].

However, in oral squamous cell carcinomas, three categories of CSC have been identified according to the expression of CD44, ESA (epidermal surface antigen) and ALDH1 (aldehyde dehydrogenase 1). The CD44^high^/ESA^low^ cells are identified as mesenchymal CSCs while the CD44^high^/ESA^high^ are epithelial. Moreover, the same team have shown that all epithelial CSCs can reconstitute tumor heterogeneity (epithelial and mesenchymal cells), while only a few mesenchymal CSCs can do the same. This heterogeneity in mesenchymal CSC population is characterized by ALDH1 expression. Indeed, CD44^high^/ESA^low^/ALDH1^high^ cells are mesenchymal CSCs able to restore tumor heterogeneity, while CD44^high^/ESA^low^/ALDH1^low^ cells cannot [130]. This could explain why very mesenchymal cells could not systematically metastasize more than strongly epithelial cells. Finally, in a model of mice spontaneous skin tumors, tumor cells with a mixed epithelial/mesenchymal phenotype can give rise to all the different tumor cell phenotypes when they are injected subcutaneously into other animals, while the more epithelial or mesenchymal cells failed [3].

### 6.6. Anticancer Treatments Targeting EMT

EMT can be promoted by multiple molecular pathways at the same time, and these pathways can be made redundant by finally converging in the same pathway. The therapeutic targeting of this phenomenon should affect different levels. The targeting of extracellular stimuli, EMT-ATFs and mesenchymal effectors seems to be a good strategy [131].

Concerning extracellular stimuli, some clinical trials are testing TGF-βRI (TGF-βReceptor I) (LY2157299) inhibitor molecule efficacy in pancreas cancers, hepatocarcinoma, glioma and glioblastoma [131,132,133,134], and anti-TGF-β antibody (Fresolimumab) efficacy in combination with radiotherapy in breast metastatic cancers [135]. Moreover, an inhibitor of the EGF receptor (erlotinib) has been approved for advanced lung cancer treatment [136]. EMT induction by hypoxia has also been targeted, at the preclinical level for now. An indirect inhibitor of HIF1α activity was shown to decrease metastasis formation in a mouse model of tumor dissemination [137].

The second class of EMT targets are the EMT-ATFs. The direct inhibition of these transcription factors is not well documented. However, several molecules can inhibit EMT-ATFs indirectly: CDK4/6 inhibitors or STAT3 inhibitors [131,138,139,140]. Immunotherapies targeting EMT-ATFs are under development. An anti-TWIST1 vaccine is effective in mice, helping to prevent tumor growth and metastasis formation in a 4T1 breast cancer model [141]. Another vaccine, targeting the EMT-ATF Brachyury, is currently under phase I and II clinical trials in different advanced cancers (NCT02383498) [142].

The third targets that could help blockade EMT are mesenchymal or epithelial effectors. Various pre-clinical studies tried to re-express the epithelial protein E-Cadherin in tumor cells. This is a difficult strategy because E-Cadherin expression has also been linked to tumor growth [131]. Re-expressing E-Cadherin can be achieved by targeting DNA methylation and histone deacetylation at its promoter during EMT [131,143]. Epigenetic mechanisms are a promising approach for cancer therapies targeting EMT. Indeed, not only E-Cadherin but many EMT-related genes are regulated epigenetically. For example, in breast, kidney or lung cancer cells, MMP9 (matrix metalloproteinase 9) and ADAM19 (Adam metallopeptidase 19) metalloproteinase expression during EMT are modulated by changes in histone methylation marks [144]. Finally, the silencing of vimentin or N-Cadherin expression could decrease the invasion and migration of tumor cell lines or mice models. The activity of these mesenchymal effectors has been modulated by specific antibodies or by chemical compounds. EMT targeting has been exhaustively reviewed by Malek et al. [131].

An EMT blockade in cancer cells can sensitize them to immunotherapies. This is why the association of these two strategies is promising and is being tested in various clinical trials. TGF-βRI inhibitors and anti-TGF-β antibodies (including anti-PD-L1 and anti-TGF-βRII fusion antibody) are currently being tested [145].

## 7. Conclusions

The increasing number of EMT studies is linked to the involvement of this phenomenon in cancers. However, EMT was discovered through Elizabeth Hay’s team’s work on chicken embryonic development. Nowadays, EMT is thought to be involved in development, wound-healing, fibrosis, and cancers. It was first considered as a binary phenomenon (called epithelial to mesenchymal transformation) transforming a full epithelial cell into a full mesenchymal one. Then, the binary nature of EMT was challenged with the identification of the inverse phenomenon: MET (mesenchymal to epithelial transition). Now, it is considered that EMT can lead to a spectrum of intermediate states between a full epithelial phenotype and a full mesenchymal one, mainly due to the work of Pastushenko et al. [2,3]. The better comprehension of this phenomenon allows us to imagine more effective cancer therapies targeting multiple features of EMT.

## Figures and Tables

**Figure 1 biomolecules-11-00782-f001:**
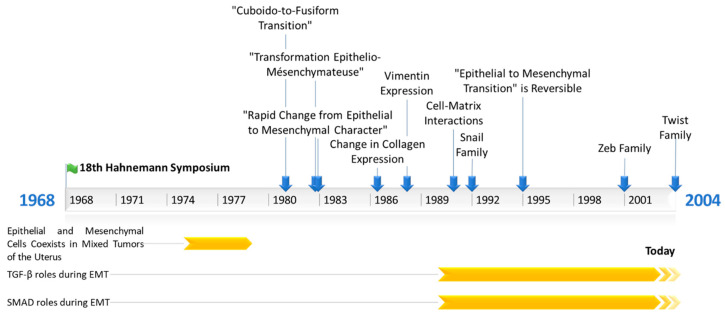
Timeline representing the main events in Epithelial to Mesenchymal Transition’s (EMT) research history. TGF-β = Transforming growth factor β; SMAD = Mothers against decapentaplegic homolog.

**Figure 2 biomolecules-11-00782-f002:**
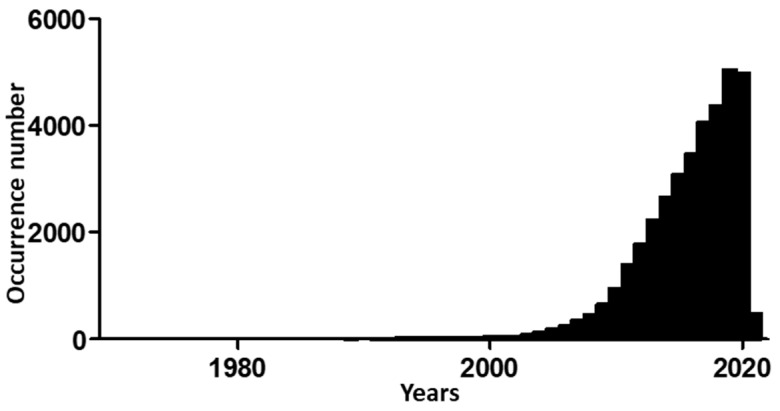
Time evolution of occurrences for “Epithelial to Mesenchymal Transition” search in PubMed database (National Center for Biotechnology Information, Bethesda, MD, USA), stopped on 27 January 2021.

**Figure 3 biomolecules-11-00782-f003:**
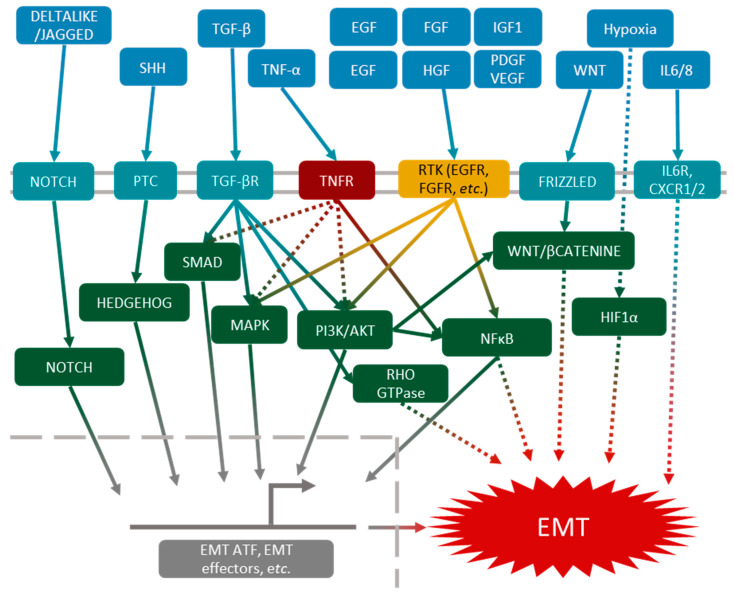
Molecular aspects of EMT induction. EMT induction can result from the activation, simultaneously or not, of one or several pathways: Notch, Hedgehog, SMAD, MAPK, PI3K/AKT, Rho GTPase, NFκB, Wnt and Hypoxia inducible factor 1α (HIF1α).

**Figure 4 biomolecules-11-00782-f004:**
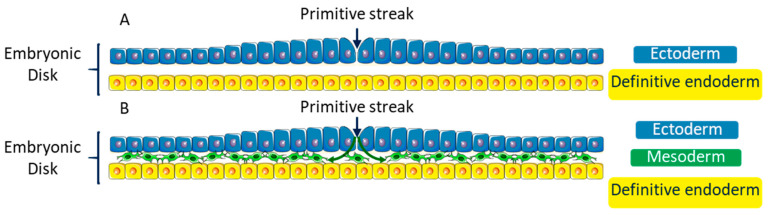
Simplified diagram of a transversal cross-section of the embryonic disk at the third week of development showing the gastrulation step. (**A**) At the beginning of the third week of development, the embryonic disk consists of two back-to-back layers: the primitive ectoderm (or epiblast) and the definitive endoderm (or hypoblast). (**B**) During the third week of embryonic development, some cells from the primitive streak of the primitive ectoderm undergo an EMT and move between the ectoderm and the endoderm to create a third mesenchymal embryonic layer: the mesoderm.

**Figure 5 biomolecules-11-00782-f005:**
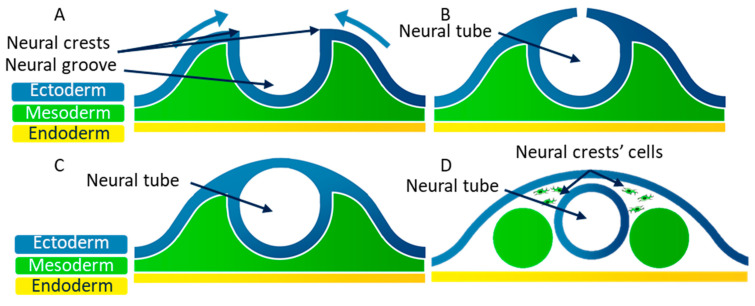
Simplified diagram of a transversal cross-section of the embryonic disk at the end of the third week of development showing the closing of the neural tube. (**A**) At the end of the third week of embryonic development, the ectoderm forms the neural groove, which creates neural crests. (**B**) Neural crest cells move to the medial edge to close the neural tube. (**C**) During the neural crests’ fusion, epithelial cells from these crests undergo an EMT to scatter in the mesoderm. (**D**) The neural tube splits from the ectoderm thanks to the scattering of the neural crest cells.

**Figure 6 biomolecules-11-00782-f006:**
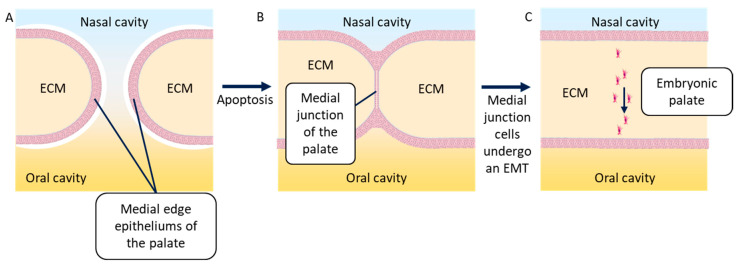
Simplified diagram of a frontal cross-section of the embryonic palatal fusion. (**A**) Medial edge epithelia of the right and left palatal shelves migrate in the direction of the medial axis of the embryo, marking out the nasal and the oral cavities. (**B**) The medial edge epithelia of the palatal shelves stick with each other to create the medial junction of the palatal shelves, thanks to the apoptosis of the cells composing the superficial layer of these epithelia. (**C**) The cells of the medial junction of the palatal shelves undergo an EMT and migrate together in the direction of the oral cavity, allowing for palatal ECM continuity. ECM = extracellular matrix.

**Table 1 biomolecules-11-00782-t001:** Genes that promote EMT or Mesenchymal to epithelial transition (MET), adapted from [27].

	Genes That Promote	
	EMT	MET
Cell surface programs	α5β1 integrin	L-CAM; E-cadherin; α6 integrin; Syndecan 1; Laminin/nidogen
Oncogenes	v-src; c-fos; v-ras; v-mos	c-met
Growth factors	TGF-β1-3; MIF; TGF-α; aFGF	HGF/SF
Other genes		wnt-1; wnt-4; pax 2; E1a

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
