# Peer review of "Epithelial to Mesenchymal Transition History: From Embryonic Development to Cancers"

_biomolecules, 2021, doi:10.3390/biom11060782_

Round 1

Reviewer 1 Report

The authors have done a good exercise compiling historical information on many of the hallmarks of EMT. From the early findings by Betty Hay, they then discuss how extracellular factors can trigger the process and then they move to the description different cell contexts where EMT occurs, both in development and cancer. Last, but not least, they elaborate on different strategies to target EMT. Overall, while many aspects have been considered, a few crucial aspects are missing: (i) the transcriptional regulation of EMT, and (ii) EMT in fibrosis. Although many reviews have been devoted to both, any historical perspective should include them as drivers of increased knowledge and understanding of the process at the cellular and molecular level.

  • Transcriptional regulation of EMT. Two main points are missing here, the description of the transcription factors that behave as EMT inducers and the regulation by microRNAs. The first is crucial to understand how the EMT is implemented upon activation by soluble factors, and the second provides a framework to understand EMT modulation. Not only because miRs attenuate (rather that suppress) gene expression, but also because they provide a beautiful mechanism of control between the epithelial and the mesenchymal phenotype by establishing reciprocal regulatory loops.
    • With respect to transcription factors, the first ones to be described were the members for the Snail family (Snail and Slug, later called Snail1 and Snail2; Smith et al., 1992; Nieto et al., 1992; Nieto et al., 1994). The latter is particularly relevant as it is the first study to show in vivo that blocking one transcription factor, the EMT is prevented. In addition, this study already suggested the connection with cancer, that was shown 6 years later to be mediated by the repression of E-cadherin transcription (Batlle et al., Nat Cell Biol 2000; Cano et al., Nat Cell Biol 2000). Interestingly, Carmen and Walter Birchmeier had previously observed the loss of E-cadherin in advanced carcinomas (Birchmeir et al., Acta Anat 1996) and Christofori´s lab had shown that E-cadherin loss was central for the transition from adenoma to carcinoma (Perl et al, Nature 1998). Additional EMT factors were later described, including Zeb (Comijn et al., Mol Cell 2001) and Twist (Yang et al., Cancer Cell, 2004). All these findings were crucial to understand the process, as these EMT factors can activate and repress multiple target genes that implement the whole program.
    • With respect to MicroRNAs and EMT, pioneer studies include the description of the miR-200 family as regulators of Zeb genes (Gregory et al., Nat Cell Biol 2008; Park et al., Genes Dev 2008) and the description of their reciprocal repression giving rise to a fundamental gene regulatory network controlling epithelial homeostasis (Burk et al., EMBO Rep 2008).
  • With respect to fibrosis, pioneer work by Eric Neilson (Iwano et al., J Clin Invest 2002), and later by Raghu Kalluri (Zeisberg et al 2007, 2008 in different journals) suggesting the role of EMT in fibrosis, plus sophisticated lineage tracing (LeBleu et al., Nat Med 2013) showing that myofibroblast do not derive from renal epithelial cells. Later on, Kalluri´s and Angela Nieto´s labs showed how EMT acts during renal fibrosis, providing an explanation for the up to then controversial data (Grande et al., Nat Med 2015; Lovisa et al., Nat Med 2015). All this work in renal fibrosis has influenced other studies in other organs. If the author prefer not to cover fibrosis, they can just adjust the title of the review to reflect development and cancer.
  • While the role of EMT in chemoresistance is well substantiated and beautifully shown in Zheng et al, 2015; Fischer et al 2015 both published in Nature (mentioned by the authors), for the role of EMT in cancer progression (breast and pancreas) and the lack of vimentin activation in breast tumours alluded to in those papers , the authors may like to read Ye et al. and also Aiello et al. both in Nature, 2017.
  • For cell plasticity in the metastatic cascade, and the regulation of stemness, in addition to Beerling et al., Cell Rep 2016 (mentioned) you could refer to previous important studies including Celia-Terrasa et al., J Clin Invest 2012, Tsai et al., Cancer Cell 2012; Ocaña et al., Cancer Cell 2012)

Reviewer 2 Report

This treatment is a pleasant and fluent reading on the historical aspects of the discovery of the importance of EMT in cell biology, both in physiological and pathological fields. I think this overview fulfills the aim of the title. What I miss is the presence of an updated description, or rather of a suitably described figure, reporting the main molecular aspects of the EMT known so far. Minor corrections are required.

Pag7 line 224 …circulating tumor cells (CTC)…..

Pag 8 line 292. In a model of spontaneous mammary mice tumors, a partial EMT, giving cells with both epithelial and mesenchymal features seems to make metastasis appearing easier, promotes collective migration of cells while full mesenchymal cells migrate alone. The sentence is not clear

Line 300. They have been described expressing to express high level of CD44 and low level of CD24 or to expressing express highly ALDH1.

Pag9 line 316

could not systematically metastasis  metastasize more than strongly epithelial cells.
